# Differential increase of hippocampal subfield volume after socio-affective mental training relates to reductions in diurnal cortisol

Sofie Louise Valk[1,2,3]\*[†], Veronika Engert[4,5][†], Lara Puhlmann[5,6], Roman Linz[5], Benoit Caldairou[7], Andrea Bernasconi[7], Neda Bernasconi[7], Boris C Bernhardt[7‡], Tania Singer[8‡]

[1]Otto Hahn Group Cognitive Neurogenetics, Max Planck Institute for Human Cognitive and Brain Sciences, Leipzig, Germany; [2]INM-7, FZ Jülich, Jülich, Germany; [3]Institute for System Neurosciences, Heinrich Heine University, Düsseldorf, Germany; [4]Institute for Psychosocial Medicine, Psychotherapy and Psychooncology, Jena University Hospital, Friedrich-Schiller University, Jena, Germany; [5]Research Group Social Stress and Family Health, Max Planck Institute for Human Cognitive and Brain Sciences, Leipzig, Germany; [6]Leibniz Institute for Resilience Research, Mainz, Germany; [7]McConnell Brain Imaging Centre, Montreal Neurological Institute and Hospital, McGill University, Montreal, Quebec, Canada; [8]Social Neuroscience Lab, Max Planck Society, Berlin, Germany

**\*For correspondence:**
s.valk@fz-juelich.de

[†]These authors contributed equally to this work
[‡]These authors also contributed equally to this work

**Competing interest:** The authors declare that no competing interests exist.

**Abstract** The hippocampus is a central modulator of the HPA-axis, impacting the regulation of stress on brain structure, function, and behavior. The current study assessed whether three different types of 3 months mental Training Modules geared towards nurturing (a) attention-based mindfulness, (b) socio-affective, or (c) socio-cognitive skills may impact hippocampal organization by reducing stress. We evaluated mental training-induced changes in hippocampal subfield volume and intrinsic functional connectivity, by combining longitudinal structural and resting-state fMRI connectivity analysis in 332 healthy adults. We related these changes to changes in diurnal and chronic cortisol levels. We observed increases in bilateral cornu ammonis volume (CA1-3) following the 3 months compassion-based module targeting socio-affective skills (*Affect* module), as compared to socio-cognitive skills (*Perspective* module) or a waitlist cohort with no training intervention. Structural changes were paralleled by relative increases in functional connectivity of CA1-3 when fostering socio-affective as compared to socio-cognitive skills. Furthermore, training-induced changes in CA1-3 structure and function consistently correlated with reductions in cortisol output. Notably, using a multivariate approach, we found that other subfields that did not show group-level changes also contributed to changes in cortisol levels. Overall, we provide a link between a socio-emotional behavioural intervention, changes in hippocampal subfield structure and function, and reductions in cortisol in healthy adults.

## eLife assessment

This **important** work examines the potential utility of socio-emotional and socio-cognitive mental training on hippocampal subfield structure and function, and cortisol levels. The authors provide **convincing** evidence that CA1-3 volume is sensitive to socio-emotional training, with changes related to function plasticity and cortisol levels. Further, the authors provide evidence of change across all subfields and training modules related to stress.

## Introduction

Stress-related disorders rank among the leading causes for disease burden world-wide (*GBD 2015 Disease and Injury Incidence and Prevalence Collaborators, 2016*), and the global stress load has increased even more dramatically in recent years (*Brooks et al., 2020*; *Robillard et al., 2020*). It is therefore essential to find ways to efficiently prevent or reduce stress (*Jorm et al., 2017*). In recent years, research has shown that contemplative mental training programs can be efficient in stress reduction (*Puhlmann et al., 2021b*; *Engert et al., 2023*; *Engert et al., 2017*; for a meta-analysis see *Khoury et al., 2015*), while simultaneously inducing brain plasticity (*Valk et al., 2017*; *Pernet et al., 2021*; *Davidson, 2016*). It is, however, still unclear which types of mental practices are most efficient in reducing stress and inducing stress-related brain plasticity. Furthermore, stress is a multi-layered construct (*Engert et al., 2018*), and most studies focused on stress-related self-reports and questionnaires (*Khoury et al., 2015*). A less investigated marker in the stress reduction context through contemplative mental training is diurnal cortisol, from which summary indices such as the cortisol awakening response (CAR), the total diurnal output and the diurnal cortisol slope are frequently investigated (*Ross et al., 2014*). The steroid hormone and glucocorticoid cortisol is the end-product of the hypothalamic-pituitary-adrenal (HPA) axis and a key player in stress regulation (for reviews see e.g., *Chrousos, 2009*; *Sapolsky, 2000a*). Moreover, cortisol is considered an important mediator of the relation between chronic stress and stress-related disease (*Adam et al., 2006*; *Miller et al., 1995*). Previous research suggests an association between hippocampal structural integrity and stress-related cortisol activity (*McEwen et al., 2016*; *McEwen, 1999*), although findings are inconclusive. To close these gaps, we here investigate the differential efficiency of three types of mental training (attention-based, socio-affective and socio-cognitive) on their ability to induce structural as well as functional plasticity of hippocampal subfields and reduce diurnal cortisol levels.

The hippocampus has a high glucocorticoid receptor density (*Herman and Cullinan, 1997*; *Herman et al., 1995*; *Jacobson and Sapolsky, 1991*; *McEwen, 2013*), making this region a target of investigations into stress-related brain changes. Having a three layered allocortex, the hippocampal formation consists of multiple subfields, or zones, starting at the subiculum (SUB) and moving inward to the hippocampus proper, the cornu ammonis (CA1-3), and dentate gyrus (CA4/DG) (*Palomero-Gallagher et al., 2020*; *Wisse et al., 2017*; *Yushkevich et al., 2015*; *DeKraker et al., 2018*). These subfields have unique microstructure (*Palomero-Gallagher et al., 2020*; *Wisse et al., 2017*; *Yushkevich et al., 2015*; *DeKraker et al., 2018*; *Paquola et al., 2020*) and participate differently in the hippocampal circuitry (*de Flores et al., 2017*), likely implicating different functional contributions (*Hodgetts et al., 2017*; *Berron et al., 2016*; *Plachti et al., 2019*; *Genon et al., 2021*; *Olsen et al., 2012*; *Vos de Wael et al., 2018*). Indeed, intrinsic functional MRI analyses have shown that the hippocampal subfields show functional correlation with a broad range of cortical regions, part of visual, control, and default functional networks (*Paquola et al., 2020*; *Vos de Wael et al., 2018*; *Bayrak et al., 2022*; *Zhong et al., 2019*; *Przeździk et al., 2019*). Hippocampal subfield volumes and associated intrinsic functional connectivity have been shown to be heritable (*Bayrak et al., 2022*; *Whelan et al., 2016*), indicating that individual variation in subfield structure and function is, in part, under genetic control. Other lines of research have reported hippocampal structure and function to be highly sensitive to contextual factors, such as stress (*McEwen, 2013*). Mediated through its dense network of glucocorticoid receptors, the hippocampus transmits the negative feedback signals of a wide range of glucocorticoid levels on HPA axis activity (*Jacobson and Sapolsky, 1991*). Through this inhibitory role on HPA axis dynamics, it is linked to emotional reactivity (*Phelps, 2004*), stress sensitivity (*McEwen, 1999*; *Bannerman et al., 2004*; *Franklin et al., 2012*; *Pruessner et al., 2010*), and causally involved in a variety of stress-related disorders (*O'Doherty et al., 2015*).

Previous brain imaging research has examined the relationship between cortisol activity and hippocampal structure and function. Most of this research measured saliva cortisol levels to gauge the diurnal cortisol profile. Thus, a reduced cortisol awakening response, the response to the anticipated demands of the upcoming day (*Fries et al., 2009*), has been associated with smaller hippocampal volume in healthy individuals (*Pruessner et al., 2007*; *Pruessner et al., 2005*; *Bruehl et al., 2009*) and different psychiatric (*Dedovic et al., 2010*; *Dedovic et al., 2009*) and metabolic (*Bruehl et al., 2009*; *Ursache et al., 2012*) conditions. In fact, the examination of patients with temporal lobe damage suggested that hippocampal integrity may be a necessary condition for the proper mounting of the CAR (*Buchanan et al., 2004*; *Wolf et al., 2005*). Next, to changes in hippocampal

**eLife digest** Too much stress is harmful to the brain and overall health, as it can lead to chronically high levels of the stress hormone cortisol. The part of the brain that regulates memory and emotions, called the hippocampus, is especially sensitive to stress because it has a high number of receptors that bind to cortisol. This connection may explain why stress can lead to memory lapses or strong emotions.

Studies have shown that mental training exercises, such as mindfulness and meditation, may change the structure of the brain and reduce stress. However, the results from these experiments have been mixed due to the variation in mental practices used by different programs. Here, Valk, Engert et al. set out to find how distinct types of mental training affect the brain, focusing on the hippocampus and cortisol levels.

The team used various magnetic resonance imaging techniques to study the hippocampus of 322 healthy adult volunteers who had undergone three months of mental training. The relationship between mental training, hippocampus size, and stress levels was complex when studying the results of each individual. However, when the results were grouped together, it revealed that volunteers who underwent training to increase empathy and compassion experienced expansion in parts of their hippocampus.

As these areas of the brain increased in size, these individuals experienced corresponding reductions in cortisol levels. But volunteers who underwent mental training focused on attention or developing perspective did not experience such increases.

These findings suggest that mental training to increase empathy and compassion alters brain structure and lowers cortisol levels. Future studies may explain how this happens, which could lead to improved mental training programs for mitigating stress.

structure, alterations in hippocampal functional connectivity have been reported to be associated with changes in cortisol levels (*Hakamata et al., 2019*; *Chang and Yu, 2019*). There is also contrary work showing associations between elevated awakening, evening, diurnal, or 24 hour cortisol levels in healthy elderly with age-related hippocampal atrophy (*Knoops et al., 2010*; *Lupien et al., 1999*; *Lupien et al., 1998*; *Sudheimer et al., 2014*) and, again, samples with psychiatric conditions (*Gold et al., 2010*; *Mondelli et al., 2010*). While such inconsistencies in previous neuroimaging work may reflect the fact that different indices of diurnal cortisol tap different facets of HPA axis regulation, the samples studied have been diverse in terms of health status, small in size, and largely cross-sectional. In addition, associations between stress and hippocampal structure and function over time are incompletely understood. Thus, longitudinal studies, such as mental training studies aimed at stress reduction that repeatedly assess both brain and cortisol release, may help to better understand the dynamic relationships between stress, cortisol, and hippocampal structure and function.

In recent years, contemplative mental training interventions, such as the mindfulness-based stress reduction (MBSR) program (*Kabat-Zinn, 1982*) or compassion-focused therapy (*Gilbert, 2014*), have gained in popularity as potential therapeutic tools to improve mental and physical health (*Goldberg et al., 2018*) and reducing stress (*Khoury et al., 2015*). These mental training interventions can have a positive impact on the practitioner's stress sensitivity, making them a suitable model to investigate the interrelationship between training-related changes in hippocampal structure, function, and cortisol output. Next to reductions in reactive measures following acute psychosocial stress induction in the laboratory (*Engert et al., 2017*), reduced subjective-psychological stress load is the most widely reported outcome (for a review, see *Khoury et al., 2015*). Evidence for lower diurnal cortisol output stems mainly from mindfulness-based interventions, notably MBSR, for which reductions in CAR and afternoon/evening cortisol levels have been reported in healthy and diseased individuals (*Brand et al., 2012*; *Carlson et al., 2007*; *Carlson et al., 2004*). Moreover, other work in the current sample has shown that hair cortisol and cortisone are reduced through mental practice (*Puhlmann et al., 2021b*). Hair cortisol measurements have been suggested to provide a window into long-term impact of cortisol exposure (*Stalder et al., 2017*). These findings are contrasted by numerous null results (for meta-analyses see: *Pascoe et al., 2017*; *Sanada et al., 2016*), possibly due to modest samples sizes and mixed effects of different training contents on stress-related processes. Furthermore, 8 weeks

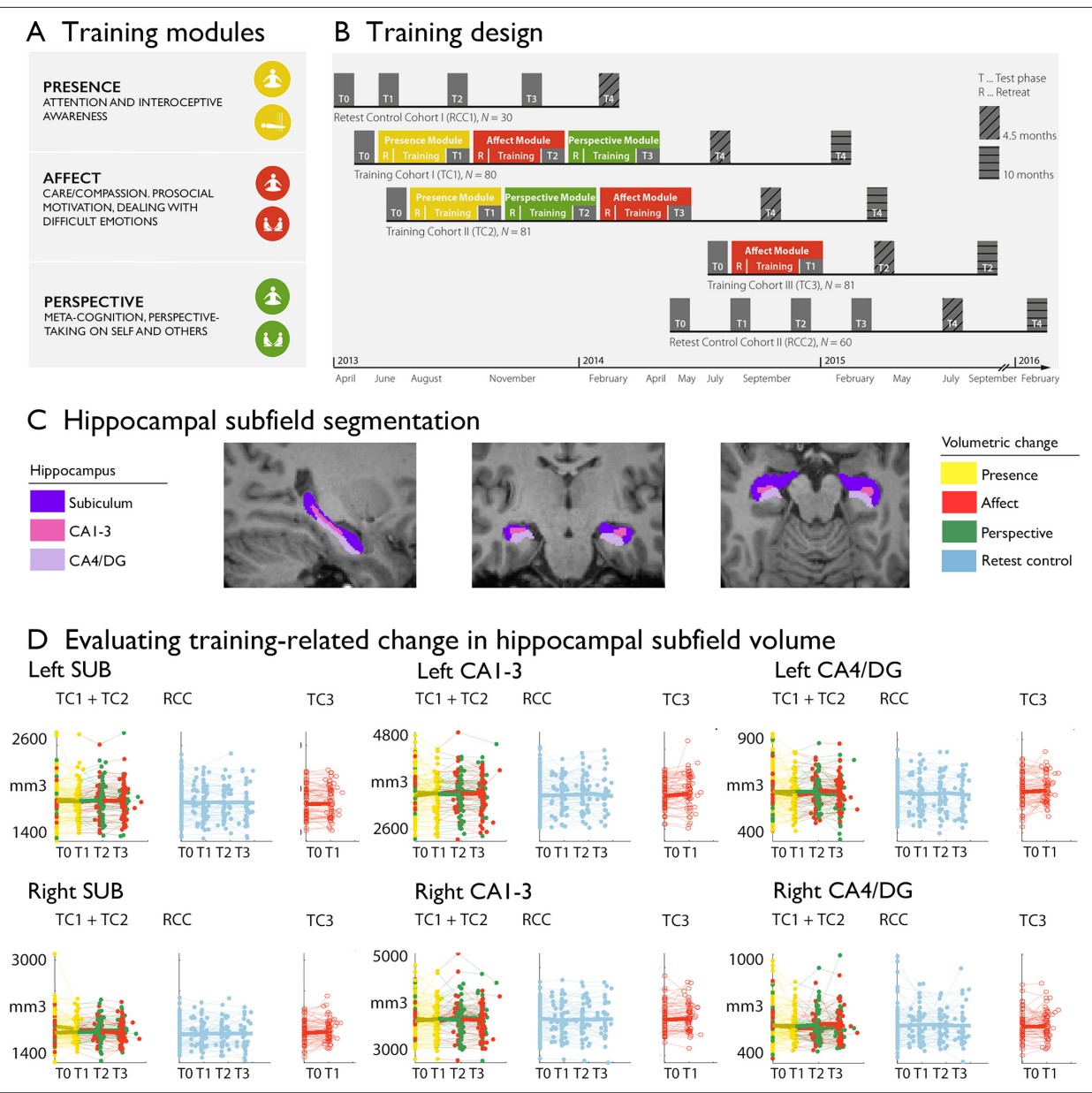

**Figure 1.** Training induced plasticity of hippocampal subfield volume. (**A**) Training modules; (**B**) Training design; (**C**) Subfield volumes in left and right hemispheres across individuals and timepoints; (**D**) Scatterplot of subfield volumes as a function of timepoints and training cohorts. Panels A and B are reproduced from Figure 1A, B of *Valk et al., 2023a*.

mindfulness programs such as MBSR and others typically cultivate different types of mental practices, making it difficult to understand which type of mental practice is most efficient in reducing different types of outcomes, including various stress-markers (see also *Puhlmann et al., 2021b*; *Engert et al., 2023*; *Engert et al., 2017*).

The current study, therefore, investigated differential effects of distinct mental training practices onto the association between changes in hippocampal subfields and underlying stress-related diurnal cortisol profiles changes in the context of a large-scale 9month mental training study, the *ReSource Project* (*Singer et al., 2016*). We explored impact of long-term exposure to stress onto hippocampal subfields as a function of mental training in a subset of individuals (*Puhlmann et al., 2021b*). Healthy participants attended three 3 months Training Modules termed *Presence* (cultivating attention and interoceptive awareness), *Affect* (cultivating compassion, prosocial motivation and dealing with difficult emotions) and *Perspective* (cultivating metacognition and perspective-taking on self and others)

**Table 1.** Sample size per timepoint.

|  | Structural MRI data | Structural and Functional MRI data |
|---|---|---|
| T0 | 288 (TC3:71) | 258 (TC3: 70) |
| T1 | 272 (TC3:68) | 238 (TC3: 64) |
| T2 | 193 | 172 |
| T3 | 190 | 181 |

(*Figure 1*). *Presence* resembles typical mindfulness-based interventions, but excludes socio-emotional or socio-cognitive practices (*Kabat-Zinn, 1982*; *Segal et al., 2002*). By contrast, *Affect* and *Perspective* target social skills through the training of either socio-emotional and motivational skills such as empathy, compassion and care (*Affect*) or socio-cognitive skills such as perspective taking on self and others (*Perspective*). In previous work, stemming from the same participant sample as examined here, we found a reduction in CAR specifically after the training of socio-affective capacities (*Engert et al., 2023*), and of acute stress reactivity after the training of socio-affective or socio-cognitive capacities (*Engert et al., 2017*). Alternatively, but also in the current sample, different types of mental practices equally reduced hair cortisol levels, a marker of long-term stress (*Puhlmann et al., 2021b*). This suggests that the content of mental training has a specific effect on daily cortisol changes but not on long-term stress levels. Our group could also show differentiable training-related changes in cortical structure and intrinsic functional organization following the three *ReSource* project Training Modules, illustrating the existence of training-related structural plasticity of the social brain (*Valk et al., 2017*; *Valk et al., 2023b*). Domain-specific changes in hippocampal subfield structure and intrinsic functional connectivity, and how these relate to mental training specific changes in stress-related diurnal cortisol output, have not yet been studied. We, therefore, examined whether module-specific changes in diurnal cortisol levels may relate to specific structural and intrinsic functional changes in different hippocampal subfields and functional resting state data.

We evaluated the longitudinal relationship between hippocampal subfield volumetry, a quantitative index of hippocampal grey matter, and studied whether volumetric changes were paralleled by subfields' resting-state functional connectivity in a large sample of healthy adults participating in the *ReSource Project* (*Singer et al., 2016*). This enabled us to evaluate training effects on hippocampal structure, function, and their associations with cortisol as a function of mental training targeting either attention-based mindfulness (*Presence*), socio-affective (*Affect*), or socio-cognitive (*Perspective*) skills. Hippocampal structure was quantified via a surface-based multi-template algorithm that has been shown to perform with excellent accuracy in healthy and diseased populations of a comparable age range as the currently evaluated cohort (*Caldairou et al., 2016*). Such a model is good to represent

**Table 2.** Reason for missing data across the study duration.
*MR incidental findings* are based on $T_0$ radiological evaluations; participants who did not meet *MRI quality control* criteria refers to movement and/or artefacts in the T1-weighted MRI; dropout details can be found in *Singer et al., 2016*; *no MRT*: due to illness / scheduling issues / discomfort in scanner; *other*: non-disclosed; *functional MRI missing:* no complete functional MRI; *functional MRI quality:* >0.3 mm movement (low quality in volume +surface).

| Reason for dropout (TC1, TC2, RCC: N=251) | $T_0$ | $T_1$ | $T_2$ | $T_3$ |
|---|---|---|---|---|
| Structural MR incidental finding | 5 | (5 based on $T_0$) | (5 based on $T_0$) | (5 based on $T_0$) |
| Structural MRI quality control | 7 | 6 | 4 | 2 |
| Dropout | 2 | 7 (2 based on $T_0$) | 9 (7 based on $T_{01}$) | 16 (9 based on $T_{012}$) |
| Medical reasons | 1 | 7 (1 based on $T_0$) | 8 (7 based on $T_{01}$) | 15 (8 based on $T_{012}$) |
| Other | 4 | 10 | 7 | 7 |
| Functional MRI missing/low QC | 29 | 30 | 21 | 9 |
| Hippocampal QC | 15 | 12 | 25 | 16 |

**Table 3.** Reason for missing data across the study duration.

*MR incidental findings* are based on $T_0$ radiological evaluations; participants who did not survive *MRI quality control* refers to movement and/or artefacts in the T1-weighted MRI; dropout details can be found in *Singer et al., 2016*; *no MRT*: due to illness / scheduling issues / discomfort in scanner; *other*: non-disclosed.

| Reason for dropout (TC3, N=81) | $T_0$ | $T_1$ |
|---|---|---|
| MR incidental finding | 3 | (3 based on $T_0$) |
| MRI quality control | 0 | 0 |
| Dropout | 0 | 3 |
| Medical reasons | 1 | 2 |
| Other | 5 | 3 |
| Functional MRI missing | 1 | 4 |
| Hippocampal QC | 1 | 2 |

different subfields in vivo, which have a differentiable structure and function (*Paquola et al., 2020*; *DeKraker et al., 2021*), and thus may show differentiable changes as a function of mental training. We expect that assessment of hippocampal sub-regions may help to accurately map circuit plasticity as a result of potential stress reduction, and to observe that changes in hippocampal structure are paralleled by changes in functional connectivity of hippocampal subfield functional networks. To model the interplay between individual-level correspondence in hippocampal and stress markers, we assessed the association of changes in hippocampal structure and function with changes in several indices of diurnal cortisol release.

## Results

We analyzed structural MRI, resting-state functional MRI, as well as cortisol-based stress markers from the large-scale *ReSource Project* (*Singer et al., 2016*). For details, see http://resource-project.org and the preregistered trial https://clinicaltrials.gov/ct2/show/NCT01833104.

In the *Resource* study, participants were randomly assigned to two training cohorts (TC1, N=80; TC2, N=81) and underwent a 9 months training consisting of three sequential Training Modules (*Presence*, *Affect*, and *Perspective*) with weekly group sessions and daily exercises, completed via cell-phone and internet platforms (*Figure 1*, *Tables 1 and 2*, *Table 3*, *Materials and Methods and Supplementary file 1 for more details*). TC1 and TC2 started their training regimen with the *Presence* module, and then underwent the latter two modules in different orders (TC1: *Affect-Perspective*; TC2 *Perspective-Affect*) to serve as active control groups for each other (*Figure 1C*). Another active control group (TC3; N=81) completed three months of *Affect* training only. Additionally, a matched test-retest control cohort did not undergo any training (RCC, N=90). All participants were examined at the end of each 3 months module ($T_1$, $T_2$, $T_3$) using 3T MRI, behavioral and peripheral-physiological measures that were identical to the baseline ($T_0$) measures.

### Change in bilateral CA1-3 volume following Affect mental training

The above design allowed us to examine whether the volume of hippocampal subfields shows increases or decreases following the distinct Training Modules. We tracked longitudinal changes in hippocampal subfield volumes using mixed-effects models (*Caldairou et al., 2016*). Excluding participants with missing or low quality structural and functional data, the sample included 86 individuals for *Presence*, 92 individuals for *Affect*, 83 individuals for *Perspective*, and 61 *active controls (Affect)* with hippocampal change scores. We included 164 change scores of *retest controls* over $T_1$, $T_2$, $T_3$. To study whether there was any training module-specific change in hippocampal subfield volumes following mental training, we compared training effects between all three Training Modules (*Presence*, *Affect*, and *Perspective*). Main contrasts were: *Presence* vs *Active control* (between subjects) and *Affect* vs *Perspective* (within subjects). Supplementary comparisons were made vs retest controls and within training groups. We observed relative increases in right cornu ammonis 1–3 (CA1-3), but not in

subiculum (SUB) or CA4 and dentate gyrus (CA4/DG) subfields, following *Affect* versus *Perspective* training (left: t=2.360, p=0.019, FDRq(q)>0.1, Cohens D=0.282; right: t=2.930, p=0.004, q=0.022, Cohens D=0.350), that could be attributed to subtle increases (p<0.05) in bilateral CA1-3 volume following *Affect* (left: t=2.495, p=0.013, q=0.08, M: 25.511, std: 130.470, CI [−1.509 52.531]; right: t=2.374, p=0.018, q>0.1, M: 40.120, std: 181.300, CI [2.573 77.666]), and subtle decreases (p<0.05) in right CA1-3 volume following *Perspective* (left: t=−1.143, p>0.1, q>0.1, M:−23.048, std: 137.810, CI [−53.139 7.043]; right: t=−2.118, p=0.035, q>0.1, M:−39.602, std: 208.470, CI [−85.122 5.917]). We did not observe differences between *Presence* and the Active control cohort, *Affect* TC3. Overall, for all hippocampal subfields, findings associated with volume increases in CA1-3 following the *Affect* training were most consistent across timepoints and contrasts (*Supplementary file 1a-g*). Moreover, associations between CA1-3 and *Affect*, relative to *Perspective*, seemed to go largely above and beyond changes in the other subfields (left: t-value: 2.298, p=0.022, Q>0.1; right: t-value: 3.045, p=0.0025, Q=0.015, see further *Supplementary file 1h*). We observed no overall change in hippocampal subfield volume following mental training of nine months (*Supplementary file 1i*). Although stereotaxic normalization to MNI space would in theory account for global sex differences in intracranial volume (ICV), we still observed sex differences in various subfield volumes at baseline. Yet, accounting for ICV did not impact our main results, suggesting changes in CA1-3 following *Affect* were robust to sex differences in overall brain volume (*Supplementary file 1j*).

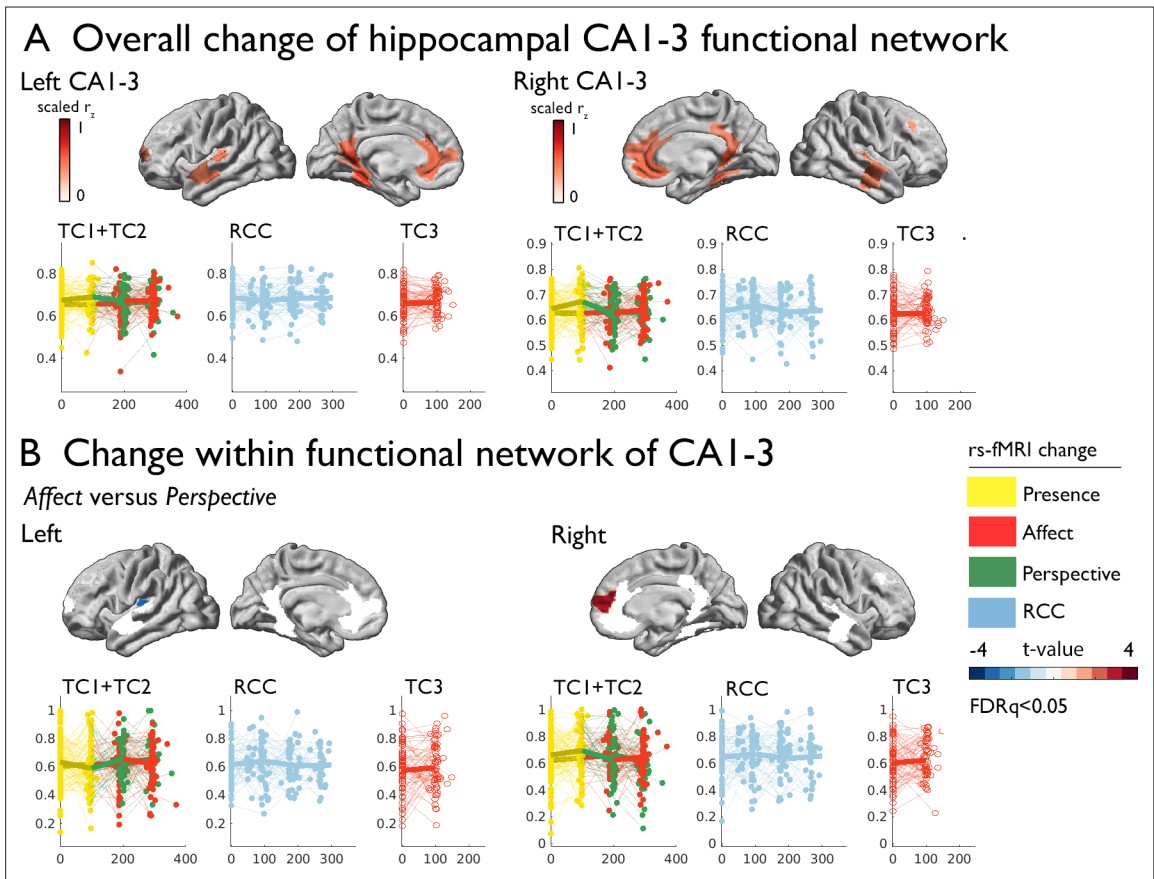

**Figure 2.** Training induced plasticity of CA1-3 functional connectivity. (**A**) *upper:* CA1-3 functional connectivity at baseline, top 10% of regions representing the CA1-3 functional network; *lower:* scatter plot visualizing change within the CA1-3 network across timepoints and groups; networks and scatters of SUB and CA4/DG are available in the supplements; (**B**) Regional change within CA1-3 functional network Affect versus Perspective (FDRq <0.05); *right:* scatter plot visualizing mean change within the CA1-3, FDRq <0.05 regions across timepoints and groups.

**Table 4.** Changes in mean CA1-3 functional network between training and active control cohorts [T0-T1] and [T1-T3].

| *Affect* TC3 vs *Presence* | LCA1-3 | RCA1-3 |
|---|---|---|
| *t-value* | 0.366 | −0.411 |
| *p- and q-value* | *p>0.1, q>0.1* | *p>0.1, q>0.1* |
| *Cohens D* | 0.052 | −0.058 |
| ***Affect* vs *Perspective*** | | |
| *t-value* | 0.137 | 2.420 |
| *p- and q-value* | p=0.891, q>0.1 | p=0.016, q=0.032 |
| *Cohens D* | 0.016 | 0.289 |

## Increased functional connectivity of CA1-3 following socio-affective versus socio-cognitive mental training

Subsequently, we studied whether volumetric change in hippocampal CA1-3 would show corresponding changes in intrinsic function following the *Affect* mental training. To probe the CA1-3 functional connectivity networks per subfield, we mapped the top 10% of normalized functional connections at baseline. Functional connectivity was strongest to medial prefrontal regions, precuneus extending to posterior cingulate, anterior temporal regions and angular gyrus (CA1-3: *Figure 2*; see *Supplementary file 1* for other subfields). Evaluating functional connectivity changes, we found that the right CA1-3 functional network showed differential changes when comparing *Affect* training to *Perspective* training (2.420, p=0.016, q=0.032, Cohens D=0.289), which could be attributed to subtle (p<0.05) decreases in right CA1-3 mean FC following *Perspective* (t=−2.012, p=0.045, q>0.1, M:−0.024, std: 0.081, CI [-0.041–0.006]), but not *Affect* training (t=1.691, p=0.092, q>0.1, M: 0.010, std: 0.098, CI [–0.01 0.031]); changes were not present when comparing *Affect* training versus retest control (*Table 4* and *Supplementary file 1k-q*). Comparing *Affect* TC3 relative to *Presence* training, we did not observe changes (*Table 4*). No other subfield showed differential changes in main contrasts within its functional network.

Exploring whether particular regions within the CA1-3 network showed alterations in intrinsic functional connectivity when comparing *Affect* to *Perspective*, we investigated connectivity changes within regions of the subfields' functional networks. Left CA1-3 connectivity showed decreases in connectivity to left posterior insula when comparing *Affect* to *Perspective* training (FDRq <0.05; t=−3.097, p=0.003, Cohens D=−0.370). On the other hand, we observed connectivity increases between right CA1-3 to right mPFC for the same contrast (FDRq <0.05; t=3.262, p=0.002, Cohens D=0.389). No other subfield's functional connectivity showed alterations when comparing *Affect* to *Perspective* or *Presence* to *Affect* TC3. These analyses indicate an overlap between volumetric increases and functional alterations when comparing changes following socio-affective mental training in CA1-3. In particular, the moderately consistent CA1-3 volume increases following *Affect* training were complemented with differential functional connectivity alterations of this subfield when comparing *Affect* to *Perspective* training.

## Association between change in subfield volume, function, and stress markers

Last, we probed whether group-level changes in hippocampal subfield CA1-3 volume would correlate with individual-level changes in diurnal cortisol indices (*Presence*: n=86; *Affect*: n=92; *Perspective*: n=81), given that the hippocampal formation is a nexus of the HPA-axis (*McEwen, 1999*). We took a two-step approach. First, we studied univariate associations between cortisol and subfield change, particularly focusing on the *Affect* module and CA1-3 volume based on increases in CA1-3 volume identified in our group-level analysis. We observed that increases in bilateral CA1-3 following *Affect* showed a negative association with change in total diurnal cortisol output (operationalized as the area under the curve with respect to ground; $AUC_g$) (*Table 5*, left: t=−2.237, p=0.028, q=0.056; right: t=−2.283, p=0.025, q=0.05), indicating that with a reduction in stress-levels as measured by $AUC_g$,

**Table 5.** Correlating change in CA1-3 subfield volume and diurnal cortisol indices in *Affect*.

| | LCA1-3 | RCA1-3 |
|---|---|---|
| CAR | –0,355, p>0.1 | –1,543, p>0.1 |
| Slope | –0,878, p>0.1 | –1,245, p>0.1 |
| AUC$_g$ | –2,237, p=0.028, q=0.056 | –2,283, p=0.025, q=0.05 |

there were increases in CA1-3 volume. *Post-hoc* analyses indicated no other subfield showed an association with AUC$_g$, or with any of the other cortisol indices, below p<0.05 (*Supplementary file 1r*). Assessing the associations between cortisol indices and the right CA1-3 subfield functional networks in *Affect* (n=92), we could not observe individual level modulation of diurnal cortisol markers and group-level effects (right CA1-3 functional network change and cortisol markers or within the PI or mPFC ROI, *Table 6* and *Supplementary file 1s*). Yet, we observed positive associations between mean functional network of *left* CA1-3 and diurnal slope (t=2.653, p=0.01, q=0.02) and AUC$_g$ (t=2.261, p=0.026, q=0.052), *Table 6* and *Supplementary file 1t*. When assessing whether particular regions within the CA1-3 network showed alterations in intrinsic functional connectivity, we observed that AUC$_g$ modulated increases in connectivity between left CA1-3 and parietal occipital area (q<0.05). These analyses extend group-level observations regarding the relation between socio-affective mental training and CA1-3 structure to the individual-level. Again, we observed some consistency in structure and function in case of CA1-3. We did not observe alterations in CA1-3 volume in relation to change in cortisol markers in *Presence* or *Perspective*. Yet, for *Presence* we observed association between slope and LCA4/DG change (t=−2.89, p=0.005, q=0.03) (*Supplementary file 1uv*). In case of intrinsic function, we also did not observe alterations in CA1-3 in relation to change in cortisol markers in *Presence* or *Perspective*, nor in other subfields (*Supplementary file 1wx*). When evaluating overall associations between diurnal cortisol change in CA1-3 volume in all modules combined, (*Presence*, *Affect*, and *Perspective*), we observed comparable patterns as for *Affect* only, further underscoring the association between cortisol markers and CA1-3 (*Figure 3B*; *Supplementary file 1y and z*). Last, we explored whether associations of subfield volume were found with levels of hair cortisol, a long-term marker of systemic cortisol exposure, in a sub-sample of N=44 participants repeatedly tested across modules (*Presence*, *Affect*, and *Perspective*), based on previous observations of domain-general effects of mental training on cortisol and cortisone (*Puhlmann et al., 2021b*). We identified consistent associations between increases in LCA1-3 volume and intrinsic function and hair cortisol decreases (volume: t=−2.574, p=0.011, q=0.022, function: t=−2.700, p=0.008, q=0.016). Exploring effects in subfields other than CA1-3 we revealed associations between right CA4/DG volume and hair cortisol (t=−3.138, p=0.002, q=0.01) and left SUB function (t=−2.890, p=0.005, q=0.03; *Figure 3B*; *Supplementary file 1za and zb*).

We employed a multivariate partial least squares method, with 1000 permutations to account for stability (*McIntosh and Lobaugh, 2004*; *Kebets et al., 2019*) and bootstrapping (100 times) with replacement, which aims to identify the directions in the predictor space that account for the most variance in changes observed, by creating latent variables. Initially, we investigated whether there was a general connection between CA1-3 subfields and cortisol changes, regardless of which Training Module produced these effects (*Figure 4*). This analysis was motivated by our observations that the bilateral CA1-3 showed increases in volume following *Affect* training and differential change between *Affect* and *Perspective* training in our resting state analyses. In a second model included structural and functional data of all subfields. Both models included all stress markers, and we regressed out age, sex and random effects of subject. We found that both models could identify significant associations

**Table 6.** Correlating change in CA1-3 subfield functional network and diurnal cortisol indices in *Affect*.

| | LCA1-3 | RCA1-3 |
|---|---|---|
| CAR | –0,476, p>0.1 | –0,425, p>0.1 |
| Slope | 2,653, p=0.009, q=0.018 | 0,773, p>0.1 |
| AUC$_g$ | 2,261, p=0.026, q=0.052 | 0,024, p>0.1 |

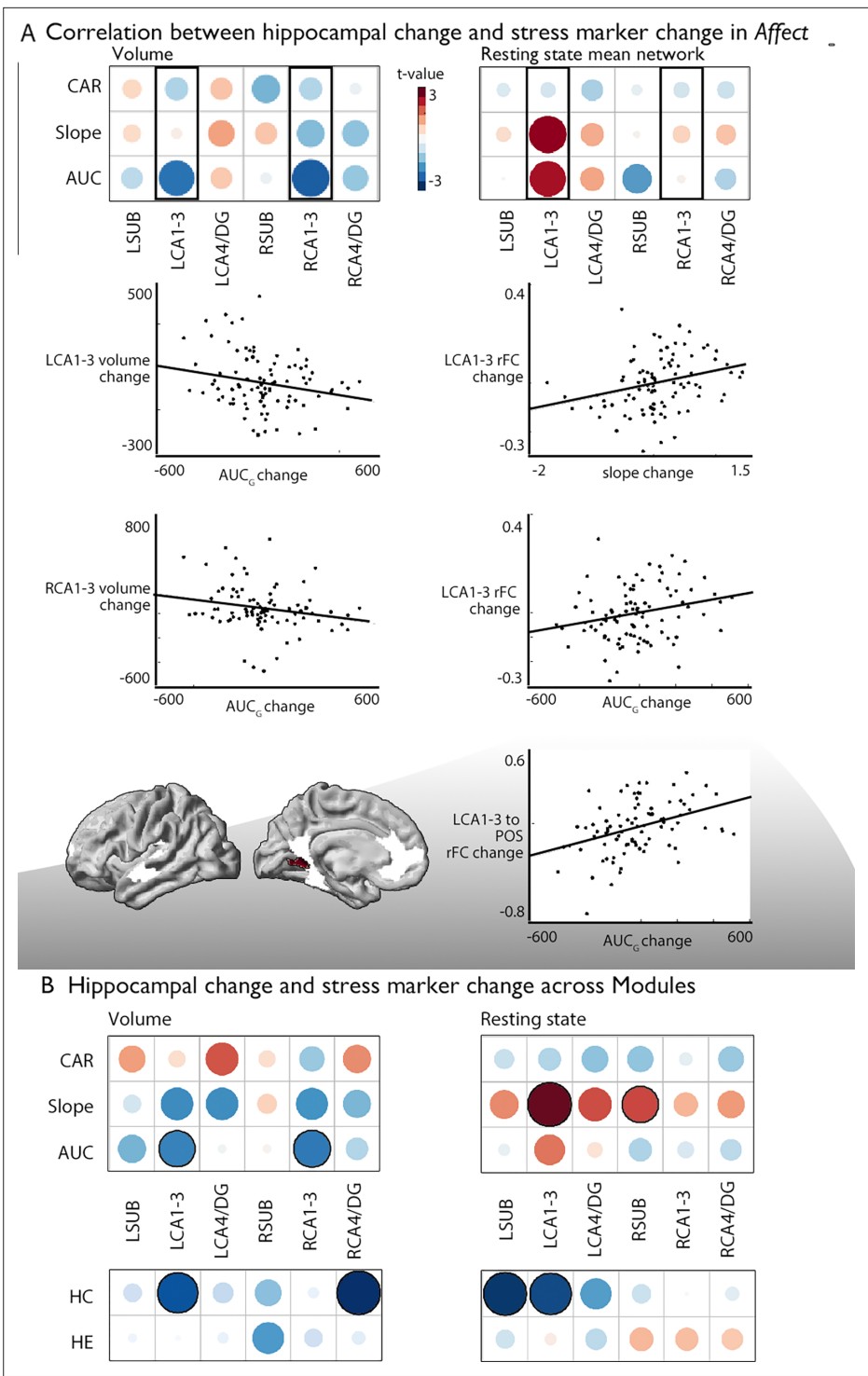

**Figure 3.** Associations between changes in structure and function of hippocampal subfield volume and markers of stress change. (**A**). *Upper left:* Correlation between hippocampal subfield volume change in *Affect* and CAR, slope, and AUC markers of stress change, *Upper right*: Correlation between hippocampal subfield intrinsic functional change in *Affect* and CAR, slope, and AUC markers of stress change, *middle:* Scatter plots visualize the correlation between volume change and cortisol marker change (below p<0.05), *bottom*: region level change within left CA1-3, FDRq <0.05. CA1-3 is the focus of this analysis based on our group-level findings and highlighted with boxes in A; (**B**). *Upper*: Overall impact of diurnal cortisol markers on hippocampal subfield volume and function over *Presence*, *Affect* and *Perspective*; *Lower*: Overall impact of hair cortisol markers on hippocampal subfield volume and function over *Presence*, *Affect* and *Perspective*.

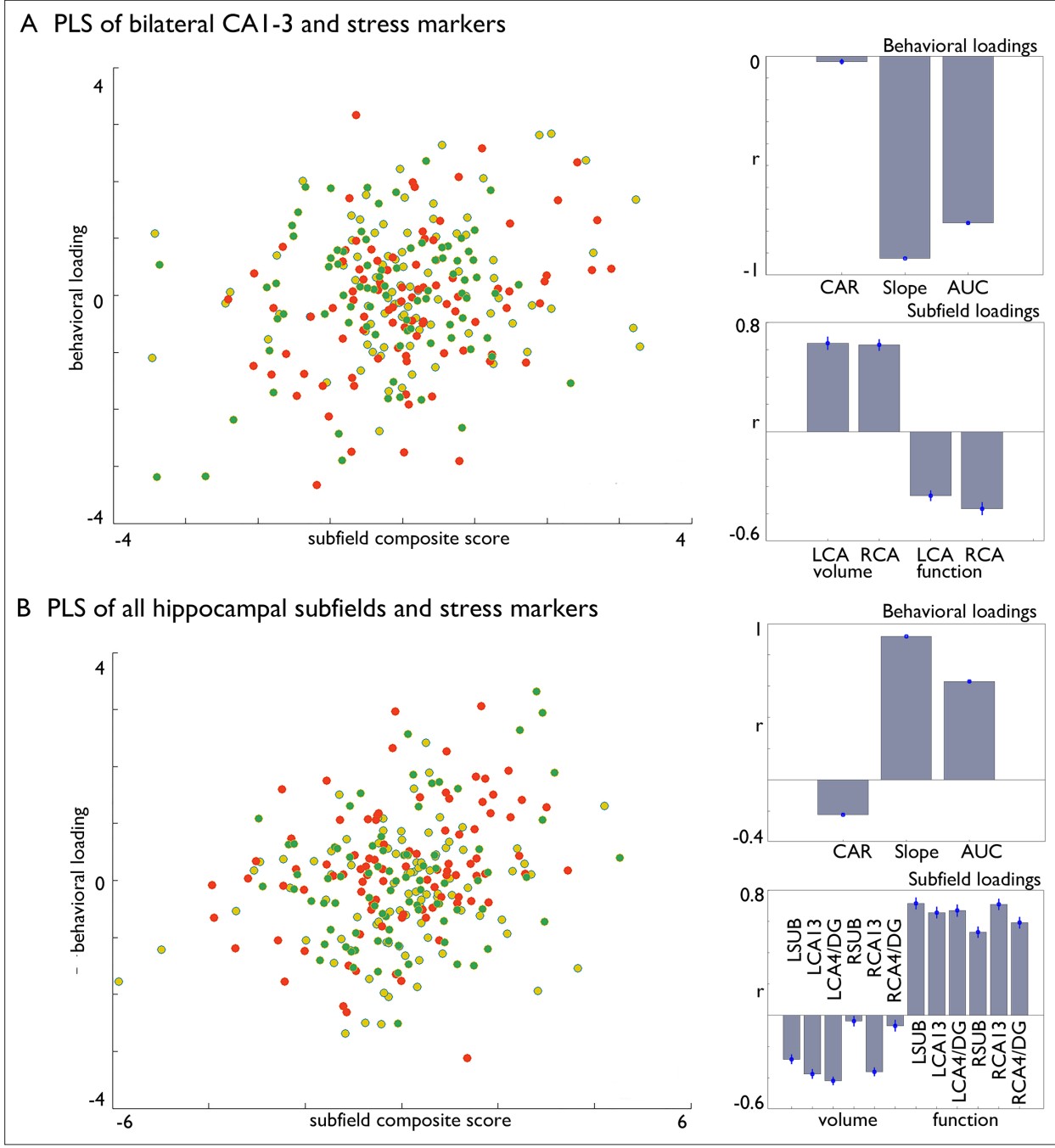

**Figure 4.** Multivariate associations between changes in structure and function of hippocampal subfield volume and markers of stress change in Affect. (**A**). Multivariate associations between bilateral CA1-3 volume and intrinsic function and stress markers. *Left*: Scatter of loadings, colored by Training Module; *Right upper*: individual correlations of stress markers; Right lower: individual correlation of subfields; (**B**). Multivariate associations between all subfields' volume and intrinsic function and stress markers. *Left*: Scatter of loadings, colored by Training Module; *Right upper*: individual correlations of stress markers; Right lower: individual correlation of subfields.

between cortisol stress markers and hippocampal plasticity (FDRq <0.05), and that in particular *Affect* showed strongest associations with the latent markers for CA1-3 (***Table 7***). Both analyses showed inverse effects of subfield structure and function in relation to stress markers and both slope and AUC changes showed strongest associations with the latent factor.

**Table 7.** Multivariate PLS analyses linking cortisol markers to hippocampal subfield volume and function.

|  | LC1 | Overall | Presence | Affect | Perspective |
|---|---|---|---|---|---|
| CA1-3 | p<0.01, 67% | r=0.20 | r=0.17 | r=0.27 | r=0.16 |
| all | p<0.01, 71% | r=0.24 | r=0.16 | r=0.30 | r=0.26 |

## Discussion

The goal of the current work was to investigate the effects of different types of mental training regimens on stress-related changes in the human hippocampus. The hippocampal formation is a highly plastic allocortical structure implicated in stress and emotional reactivity (*Chrousos, 2009*; *Sapolsky, 2000a*; *McEwen, 1999*). In this study, we used automated segmentation to examine if the volumes of hippocampal subfields (SUB, CA1-3, CA4/DG) change in a large healthy sample over a 9-month longitudinal mental training study, the *ReSource* project (*Singer et al., 2016*). We investigated whether three different interoceptive and social mental Training Modules could lead to changes in hippocampal subfield volume. Additionally, we explored if these changes were associated with alterations in both intrinsic brain function and stress-related physiological changes, as indicated by shifts in diurnal and hair cortisol levels due to training.

When comparing the differential efficiency of the 3 months mental Training Modules *Presence, Affect,* and *Perspective* against each other and a retest-control group on hippocampal subfield structure, we observed consistent increases in bilateral CA1-3 volume following socio-emotional *Affect* training relative to socio-cognitive *Perspective* training and no training in retest controls. Moreover, alterations in structure were mirrored by changes in functional connectivity of right CA1-3 following *Affect* versus *Perspective* training. In particular, we observed relative increases of functional connectivity between right CA1-3 and mPFC, and decreases between left CA1-3 and posterior insula, mainly driven by changes in connectivity following *Perspective* training. Evaluating training-related changes in diurnal cortisol output (cortisol awakening response, total diurnal output and diurnal slope), we observed that bilateral CA1-3 volume increases correlated with decreases in total diurnal cortisol output (assessed as the area under the curve with respect to ground, $AUC_g$, sampled on 10 occasions over two consecutive days). Intrinsic connectivity of CA1-3 following *Affect* showed a positive association with left CA1-3 network change and diurnal slope and total diurnal cortisol output, where the latter associated with increased connectivity between left CA1-3 and parietal-occipital area. Interestingly, these associations were similar when combining Training Modules, suggesting the association between CA1-3 and diurnal cortisol markers is present irrespective of training content. Moreover, we additionally observed consistent associations between left CA1-3 and hair cortisol, a systemic marker of chronic stress, across trainings in a sub-sample of the current study. Finally, through conducting multivariate analysis, we once more noticed associations between changes in CA1-3 volume and functional adaptability and alterations in stress levels, particularly prominent within the Affect Module. Integrating all subfields into a unified model highlighted a distinct significance of CA1-3, although for the left hemisphere, we observed a more diverse range of contributions across subfields. In summary, we establish a connection between a socio-emotional behavioral intervention, shifts in hippocampal subfield structure and function, and decreases in cortisol levels among healthy adults.

Our longitudinal, multi-modal approach could thus show that compassion-based mental training alters CA1-3 structure. Second, training-based increases in CA1-3 volume related to decreases in total diurnal cortisol release, suggesting that mental training and CA1-3 volume changes are linked to cortisol release. The results regarding changes in functional connectivity profiles were not as straightforward, but they did reveal a distinction between *Affect* and *Perspective* mental training in the CA1-3 region. Moreover, CA1-3 intrinsic functional change was associated with changes in diurnal cortisol slope and release, and long-term cortisol exposure. While the experimental nature of our training study allows concluding that CA1-3 structure changed as a function of *Affect* training, and that individual differences in CA1-3 structural change corresponded to cortisol release change, we cannot make any claims about which training-induced change caused the other. Thus, it is possible that, owing to the *Affect* module, the activation of emotion/motivation-related functional processes is key to reducing the daily stress load and associated cortisol release (*Klimecki et al., 2013*; *McCall and*

*Singer, 2012*). Such reduction in cortisol levels may then explain the observed downstream brain alterations. According to this interpretation, changes in CA1-3 volume may come secondary to stress reduction and consequently alterations in cortisol release following compassion training. Alternatively, emotion/compassion training may specifically targets CA volume and function, and, as per its role as the central break of the HPA axis, improves its capacity to inhibit cortisol release. This explanation could explain the lack of *average* diurnal cortisol (*i.e.,* AUC$_g$) change following *Affect* training per se (*Engert et al., 2023*), as it may be relevant for individual variations in brain change and thus be more difficult to detect based only on average change per module. In sum, it is likely that observed alterations in hippocampal structure and function, as well as their associations with diurnal cortisol change, are not explained by a single mechanism, but rather result as a combination of different factors. This interpretation is also supported by our multivariate observations. For example, given the anatomical and functional complexity of the hippocampal formation (*Yushkevich et al., 2015*; *Genon et al., 2021*) as well as the multifaceted cognitive processes underlying stress, it seems plausible that our observations are an emergent effect of multiple, cognitively distinct, functional sub-processes. Thus, future studies may directly test the potential specificity of the interrelationship between stress on the hippocampus using further targeted measurements.

The observed increases in CA1-3 volumes following socio-affective mental training were small-sized effects. However, findings were consistent when independently assessing the left and right hippocampus subfields. In particular, we observed that increases in CA1-3 volume after *Affect* training corresponded to a decrease in total diurnal cortisol as well as hair cortisol output. These results can be interpreted in line with the mainly inhibitory role of the hippocampus in stress regulation (*Herman and Cullinan, 1997*; *Herman et al., 1995*; *Jacobson and Sapolsky, 1991*; *Herman et al., 2005*). Specifically, the hippocampus is involved in the negative feedback inhibition of the HPA axis. Mineral- and glucocorticoid receptors are present in abundance in hippocampal neurons, from where they transmit the negative feedback signals of a wide range of glucocorticoid levels on HPA axis activity (*Jacobson and Sapolsky, 1991*). The extremely high numbers of mineral- and glucocorticoid receptors make the hippocampus a prominent target for the neurotoxic effects of glucocorticoids (*Sapolsky, 2000b*; *Sapolsky et al., 1985a*; *Sapolsky and Pulsinelli, 1985b*). In particular the CA1 may be susceptive to stress-based environmental effects due to synaptogenesis associated with NR2B subunits of glutamate receptors (NMDAR) (*Coultrap et al., 2005*). Along these lines, sustained exposure to high glucocorticoid levels was shown to relate to calcium influx, and may produce CA3 pyramidal neuronal damage, which has been reported in rodents and tree shrews (*McKittrick et al., 2000*; *Magariños et al., 1998*; *Magariños and McEwen, 1995*). Next to demonstrating a consistent relationship between total daily cortisol output and hippocampal structure, the absence of univariate findings for cortisol awakening response (CAR), diurnal slope or hair cortisone levels may a divergence in the sensitivity of alternative cortisol-based stress markers to structural neuroimaging markers. It is of note that the current work relies on a segmentation approach of hippocampal subfields including projection to MNI template space, an implicit correction for total brain volume through the use of a stereotaxic reference frame. Some caution for this method may be warranted, as complex hippocampal anatomy can in some cases lead to over- as well as underestimation of subfield volumes, as well as subfield boundaries may not always be clearly demarcated (*Wisse et al., 2021*). Future work, studying the hippocampal surface at higher granularity, for example though unfolding the hippocampal sheet (*Vos de Wael et al., 2018*; *DeKraker et al., 2021*; *DeKraker et al., 2022*; *Bernhardt et al., 2016*), may further help with both alignment and identification of not only subfield-specific change but also alterations as a function of the hippocampal long axis, a key dimension of hippocampal structural and functional variation that was not assessed in the current work (*Genon et al., 2021*; *Vogel et al., 2020*).

Structural MRI findings were complemented by the separate assessment of task-free ("resting-state") functional connectivity networks. Identification of networks that show coupled spontaneous brain activity through resting-state fMRI is currently considered an effective approach to study whole-brain functional connectivity (*Smith et al., 2011*; *Biswal et al., 1995*; *Greicius et al., 2003*). In the current cohort, we could demonstrate widespread patterns of hippocampal functional connectivity to mesiotemporal, lateral temporal, together with anterior as well as posterior midline regions, lateral temporo-parietal, and dorsolateral prefrontal cortices - a pattern in excellent accordance to previous studies probing hippocampal functional connectivity at rest in healthy populations (*Paquola et al., 2020*; *Vos de Wael et al., 2018*; *Przeździk et al., 2019*; *Buckner and Vincent, 2007*; *Vincent et al.,*

2006), and outlining 'mesiotemporal' components of default-mode networks (*Andrews-Hanna et al., 2010*; *Yeo et al., 2011*). Assessing modulations of connectivity by mental training, we could provide independent, yet weak, support for a specific relationship of the socio-affective *Affect* training, relative to socio-cognitive *Perspective* Training Modules, on hippocampal network embedding. In particular, we observed an increased functional integration of the right CA1-3 with medial prefrontal cortical regions (mPFC) in individuals following *Affect* relative to *Perspective* training. Studies in rats and non-human primates have demonstrated a high density of glucocorticoid receptors in the mPFC (*Sánchez et al., 2000*; *Ahima and Harlan, 1990*). Accordingly, the mPFC, like the hippocampus, was shown to play a key role in HPA-axis regulation (*Dedovic et al., 2009*; *Herman et al., 2005*; *Radley and Sawchenko, 2011*; *Diorio et al., 1993*). In a previous positron emission tomography study, glucose metabolism in the mPFC was negatively associated with acute stress-induced salivary cortisol increases; notably, the authors observed a negative metabolic coupling between mPFC areas and the mesiotemporal lobe (*Kern et al., 2008*). In related work on isocortical changes in structure and intrinsic function following the *ReSource* training, we have observed structural changes in insular, opercular and orbitofrontal regions following *Affect* training (*Valk et al., 2017*; *Valk et al., 2023b*). At the same time, we observed little change in large-scale functional organization, relative to changes observed following *Presence* and *Perspective* training. Previous work has implicated the hippocampal formation at the nexus of multiple large-scale networks and cortical organization (*Paquola et al., 2020*; *Vogel et al., 2020*). Indeed, it may be that particular changes in the CA1-3 are central in coordinating the signal flow within the hippocampal complex, coordinating the balance between large-scale association networks in the iso-cortex (*Paquola et al., 2020*). Integrating this with our empirical observation of *Affect* training taking up a regulatory or stabilizing functional role, relative to *Perspective* and *Presence* training, it is possible that such alterations are orchestrated by adaptive processes (*McEwen et al., 2015*). Future work may be able to further disentangle the causal relationship between iso- and allo-cortical structure and function, and the role of specific hippocampal subfields.

Using univariate approaches, we could observe that training-induced HC volume increases following socio-affective mental training overlapped with reductions in cumulative diurnal cortisol release. Additionally, we observed functional connectivity decrease between left CA1-3 and parietal-occipital area in individuals showing reduced diurnal cortisol release and overall connectivity decreases of left CA1-3 relating to reductions in diurnal cortisol slope. Importantly, these associations could be found also when including *Presence* and *Perspective* in our analysis, suggesting of a domain-general relationship between diurnal cortisol alterations and CA1-3 volume and function. In line with our observations in univariate analysis, we found multivariate associations between hippocampal subfield volume, intrinsic function and cortisol markers. Again, the contribution of volume and intrinsic function was inverse. This may possibly relate to the averaging procedure of the functional networks. Combined, outcomes of our univariate and multivariate analyses point to an association between change in hippocampal subfields and stress markers, and that these changes, at the level of the individual, ultimately reflect complex interactions within and across hippocampal subfields and may capture different aspects of diurnal stress. Future work may more comprehensively study the plasticity of the hippocampal structure, and link this to intrinsic functional change and cortisol to gain full insights in the specificity and system-level interplay across subfields, for example using more detailed hippocampal models (*DeKraker et al., 2022*). Incorporating further multivariate, computational, models is needed to further unpack and investigate the complex and nuanced association between hippocampal structure and function, in particular in relation to subfield plasticity and short and long-term stress markers. In line with our multivariate observations, in other work from the *ReSource* study we observed mixed specificity of stress-reducing effects as a function of mental training. For example, both social modules, that is the *Affect* and *Perspective* trainings, reduced acute cortisol reactivity to a psychosocial stressor (*Kirschbaum et al., 1993*), which is considered a dynamic state of HPA axis activity (*Engert et al., 2017*). Regarding the CAR, only *Affect* training was able to reduce this dynamic cortisol response to awakening, known to reflect anticipatory stress (*Engert et al., 2023*). Lastly, regarding hair cortisol, a long-term measure of systemic stress, all Training Modules were shown to be equally effective in stress reduction over a training period of three to nine months (*Puhlmann et al., 2021b*). In our work we observed a consistent association between left CA1-3 volume and functional increases and hair cortisol decreases, hinting at a potential relationship between CA1-3 and both short-term and long-term stress level changes.

Overall, different types of mental training result in stress reduction (e.g. *Puhlmann et al., 2021b*; *Engert et al., 2023*; *Engert et al., 2017*). In a recent paper we argue that the variable pattern of mental training effects on different cortisol indices may be explained by the functional roles of these indices (*Engert et al., 2023*). Thus, indices reflecting dynamic HPA axis properties, such as acute stress reactivity and the CAR, were suggested to change with *Affect* and *Perspective* training (also see *Engert et al., 2017*). Hair cortisol as a marker of cumulative stress load likely reflecting the low-grade and continuous strain inherent to daily hassles (*Almeida, 2005*; *DeLongis et al., 1982*; *Lazarus and Folkman, 1984*), was contrarily suggested to change independent of training type (also see *Puhlmann et al., 2021b*). The current findings do not necessarily contradict this reasoning, due to differences in interpretation of group-level and individual-level changes. Although we observed that CA1-3 volume was selectively increased by socio-affective mental training at the group level, and that individual differences in CA1-3 volume increase within the *Affect* module correlated with reduced diurnal cortisol release, the pattern linking bilateral CA1-3 volume increases with reduced diurnal cortisol release was also present when all modules were combined. Similarly, in follow-up analysis on functional alterations of hippocampal subfields, we could observe group-level increases in connectivity to mPFC for right, but not left, CA1-3, when comparing socio-affective and socio-cognitive training. Though right CA1-3 group-level changes did not link to individual level change in cortisol markers following *Affect* training, individual level changes in left CA1-3 corresponded to changes in cortisol markers, again following *Affect* but also across all practices combined. Thus, we cannot at this point derive a consistent pattern of how mental training influences different indices of cortisol activity, yet we do find a consistent change in CA1-3 following *Affect* training, and observe domain-general patterns of change associations between CA1-3 and cortisol markers, indicating CA1-3 may play a central role within the context of *Affect* training and diurnal stress reduction based on univariate analysis. Although the univariate examination of changes specific to modules in volume and connections within the *Affect* Module presents how changes in cortisol align with group-level rises in CA1-3 volume, the multivariate analysis extended this observation through considering individual-level alterations not discernible at the group level through a data-driven method. These results generally corresponded with observations at the group level but offer additional insights into specificity, and hint at system-level alterations. Lastly, from a mechanistic viewpoint, we hypothesize that *Affect* training stimulates emotion-motivational (reward) systems associated with positive affect (*Klimecki et al., 2013*; *McCall and Singer, 2012*), and regulated by oxytocin and opiates (*Depue and Morrone-Strupinsky, 2005*; *Nelson and Panksepp, 1998*). Since these neuropeptides are also involved in stress regulation (*Carter, 2014*; *Drolet et al., 2001*), they could be considered to provide a double hit, and prime candidates to mediate hippocampal volume increase and stress reduction in particular following compassion-based practice, yet also present following other practices.

It is of note that non-adherence to saliva sampling in ambulatory settings has been shown to exert a significant impact on the resulting cortisol data (*Kudielka et al., 2007*; *Kudielka et al., 2003*) and that the present data does not fully conform to the recently provided consensus guidelines on the assessment of the CAR (*Stalder et al., 2022*; *Stalder et al., 2016*), which were published after the conception of our study. Most importantly, we did not employ objective measures for the verification of participants' sampling times. Hence, diurnal cortisol data have to be treated with some caution since the possibility of non-adherence-related confounding cannot be excluded (*Kudielka et al., 2007*; *Kudielka et al., 2003*; *Stalder et al., 2022*; *Stalder et al., 2016*). We nevertheless addressed the issue of non-adherence through an experience sampling approach based on mobile phones handed out to our participants. As shown by the relatively low proportion of missing data, these devices may have boosted adherence by reminding participants of a forthcoming sampling time-point.

To conclude, using a longitudinal model, we investigated how different types of mental training differentially result in changes in hippocampal subfield volume, resting-state functional networks, and stress-related markers of diurnal cortisol and hair cortisol. We find that only the 3 months *Affect* training module cultivating compassion and care, rather than attention-based (*Presence*) or socio-cognitive (*Perspective*) training, related to an increase of hippocampal CA1-3 subfield volume, with corresponding alterations in functional connectivity and a reduction in total diurnal cortisol output. Across analyses we observed consistent alterations between cortisol change and CA1-3 volume and function, pinpointing this region as a potential target for further investigations on stress and the human brain. Lastly, our multivariate analyses also point to a circuit level understanding of latent

diurnal stress scores. Our results may be informative for the development of targeted interventions to reduce stress, and inspire the update of models on the role of different hippocampal formations for human socio-emotional and stress-related processes.

## Methods

The specifics on the experimental design are the similar to related works in the same sample (*Valk et al., 2017*; *Trautwein et al., 2020*). They are provided again here for completeness.

### Participants

We recruited a total of 332 healthy adults (197 women, mean ± SD = 40.7±9.2 years, 20–55 years), in the winters of 2012/2013 and 2013/2014. Participant eligibility was determined through a multi-stage procedure that involved several screening and mental health questionnaires, together with a phone interview [for details, see *Singer et al., 2016*]. Subsequently, a face-to-face mental health diagnostic interview with a trained clinical psychologist was carried out. The interview included a computer-assisted German version of the Structured Clinical Interview for DSM-IV Axis-I disorders, SCID-I DIA-X (*Wittchen and Pfister, 1997*), and a personal interview, SCID-II, for Axis-II disorders (*Wittchen et al., 1997*; *First et al., 2012*). Participants were excluded if they fulfilled criteria for: (i) an Axis-I disorder within the past two years, (ii) schizophrenia, psychotic disorders, bipolar disorder, or substance dependency, or (iii) an Axis-II disorder at any time in their life. Participants taking medication influencing the HPA axis were also excluded. None of the participants had a history of suffering from neurological disorders or head trauma, based on an in-house self-report questionnaire completed prior to the neuroimaging investigations. Included participants furthermore underwent a diagnostic radiological evaluation to rule out the presence of mass lesions (e.g. tumors, vascular malformations). The study was approved by the Research Ethics Committees of University of Leipzig (#376/12-ff) and Humboldt University in Berlin (#2013–02, 2013–29, 2014–10), and all participants provided written informed consent prior to participation. The study was registered with the Protocol Registration System of ClinicalTrials.gov under the title 'Plasticity of the Compassionate Brain' with the Identifier: NCT01833104. For more details on recruiting and sample selection, please see *Singer et al., 2016*.

### Sample size estimation and group allocation

Overall, 2595 people signed up for the *ReSource* study in winter 2012/2013. Of these individuals, 311 potential participants met all eligibility criteria. From the latter group, 198 were randomly selected as the final sample. Participants were selected from the larger pool of potential participants and assigned to cohorts using bootstrapping without replacement, creating cohorts that did not differ (omnibus test p<0.1) in demographics (age, gender, marital status, income, and IQ) or self-reported traits (depression, empathy, interoceptive awareness, stress level, compassion for self and others, alexithymia, general mental health, anxiety, agreeableness, conscientiousness, extraversion, neuroticism, and openness). Seven participants dropped out of the study after assignment but before data collection began, leaving 30 participants in RCC1, 80 in TC1, and 81 in TC2.

2144 people applied for the second wave of the study in winter 2013/2014. Of these people, 248 potential participants met all the eligibility criteria. From the latter pool, 164 were then randomly selected as the final sample. Participants were selected from the larger pool of potential participants and assigned to cohorts using bootstrapping without replacement, creating cohorts that did not differ significantly (omnibus test, p>0.1) from the Winter 2012/2013 cohorts or from one another in demographics (age, gender, marital status, income, and IQ) or self-reported traits (depression, empathy, interoceptive awareness, stress level, compassion for self and others, alexithymia, general mental health, anxiety, agreeableness, conscientiousness, neuroticism, and openness). The control cohorts (RCC1, RCC2, and RCC1&2) were significantly lower in extraversion than TC3; participants in the control cohorts were also more likely to have children than participants in TC3. Twenty-three participants dropped out of the study after assignment but before data collection began, leaving 81 participants in TC3 and 60 in RCC2. See further (*Singer et al., 2016*).

## ReSource training program

In the *ReSource Project*, we investigated the specific effects of commonly used mental training techniques by parceling the training program into three separate modules (*Presence*, *Affect,* and *Perspective*). Participants were selected from a larger pool of potential volunteers by bootstrapping without replacement, creating cohorts not differing significantly with respect to several demographic and self-report traits (*Singer et al., 2016*). Each cultivated distinct cognitive and socio-affective capacities (*Pascoe et al., 2017*). Participants were divided in two 9-month training cohorts experiencing the modules in different orders, one 3-month *Affect* training cohort and one retest control cohort. In detail, two training cohorts (TC1, TC2) started their training with the mindfulness-based *Presence* module. They then underwent *Affect* and *Perspective* modules in different orders thereby acting as mutual active control groups. To isolate the specific effects of the *Presence* module, a third training cohort (TC3) underwent the 3-month *Affect* module only (*Figure 1B*).

As illustrated in *Figure 1A*, the core psychological processes targeted in the *Presence* module are attention and interoceptive awareness, which are trained through the two meditation-based core exercises Breathing Meditation and Body Scan. The *Affect* module targets the cultivation of social emotions such as compassion, loving kindness and gratitude. It also aims to enhance prosocial motivation and dealing with difficult emotions. The two core exercises of the *Affect* module are Loving-kindness Meditation and Affect Dyad. In the *Perspective* module participants train meta-cognition and perspective-taking on self and others through the two core exercises Observing-thoughts Meditation and Perspective Dyad. The distinction between *Affect* and *Perspective* modules reflects research identifying distinct neural routes to social understanding: One socio-affective route including emotions such as empathy and compassion, and one socio-cognitive route including the capacity to mentalize and take perspective on self and others (for details on the scientific backbone of this division see: *Singer et al., 2016*).

The two contemplative dyads are partner exercises that were developed for the *ReSource* training (*Kok and Singer, 2017*). They address different skills such as perspective taking on self and others (*Perspective* dyad) or gratitude, acceptance of difficult emotions and empathic listening (*Affect* dyad), but are similar in structure (for details see: *Singer et al., 2016*). In each 10 min dyadic practice, two randomly paired participants share their experiences with alternating roles of speaker and listener. The dyadic format is designed to foster interconnectedness by providing opportunities for self-disclosure and non-judgmental listening (*Singer et al., 2016*; *Kok and Singer, 2017*). Our recommendation was to train for a minimum of 30 min (e.g. 10 min contemplative dyad, 20 min classic meditation) on five days per week.

## MRI acquisition

MRI data were acquired on a 3T Siemens Magnetom Verio (Siemens Healthcare, Erlangen, Germany) using a 32-channel head coil. Structural images were acquired using a T1-weighted 3D-MPRAGE sequence (repetition time [TR]=2300ms, echo time [TE]=2.98ms, inversion time [TI]=900ms, flip angle = 7°; 176 sagittal slices with 1 mm slice thickness, field of view [FOV]=240 × 256 mm$^2$, matrix = 240 × 256, 1×1 × 1 mm$^3$ voxels). We recorded task-free functional MRI using a T2*-weighted gradient EPI sequence (TR = 2000ms, TE = 27 ms, flip angle = 90°; 37 slices tilted at approximately 30° with 3 mm slice thickness, FOV = 210 × 210 mm$^2$, matrix = 70 × 70, 3×3 × 3 mm$^3$ voxels, 1 mm gap; 210 volumes per session). During the functional session, participants were instructed to lie still in the scanner, think of nothing in particular, and fixate a white cross in the center of a black screen.

## Structural MRI analysis: Hippocampal subfield volumetry

Based on the available high-resolution T1-weighted images subiculum (SUB), CA1-3, and CA4/DG were segmented using a patch-based algorithm in all participants individually (see further *Caldairou et al., 2016*). Shortly, this procedure uses a population-based patch normalization relative to a template library (*Kulaga-Yoskovitz et al., 2015*), providing good time and space complexity. In previous validations work, this algorithm has shown high segmentation accuracy of hippocampal subfields (*Caldairou et al., 2016*), and in detecting hippocampal subfield pathology in patients with epilepsy (*Bernhardt et al., 2016*). It was furthermore demonstrated that these representations can be used to probe sub-regional functional organization of the hippocampus (*Vos de Wael et al., 2018*; *Bayrak et al., 2022*). Hippocampal volumes were estimated based on T1w data that were linearly registered to MNI152

using FSL flirt (http://www.fmrib.ox.ac.uk/fsl/), such that intracranial volume was implicitly controlled for.

As previously reported (*Puhlmann et al., 2021a*), for successful hippocampus segmentations, an initial quality check was conducted by two independent raters, R.L. and L.P. Both raters were blind to participant characteristics including age, sex, and training or control group. In short, each segmentation was rated for quality on a scale of 1–10, with points being subtracted depending on the severity of detected flaws. One point was subtracted for minor flaws, for example part of a segmentation extends slightly beyond the hippocampal boundary, or does not cover a small aspect of the hippocampal formation. Two points were subtracted for medium flaws, for example gaps between subfield segmentations. Finally, major flaws immediately qualified for resampling, and included for example one or more subfield segmentations being clearly misplaced. Given a minimum of 70% inter-rater reliability, segmentation ratings were then averaged and evaluated, with scores of 5 and lower qualifying for reprocessing with the algorithm. Following this second round of processing, segmentations were rated again. Any remaining segmentations with average scores lower than 5 were excluded from the analysis.

## Task-free functional MRI analysis: Hippocampal connectivity

Processing was based on DPARSF/REST for Matlab [http://www.restfmri.net (*Chao-Gan and Yu-Feng, 2010*)]. We discarded the first five volumes to ensure steady-state magnetization, performed slice-time correction, motion correction and realignment, and co-registered functional time series of a given subject to the corresponding T1-weighted MRI. Images underwent unified segmentation and registration to MNI152, followed by nuisance covariate regression to remove effects of average WM and CSF signal, as well as 6 motion parameters (3 translations, 3 rotations). We included a *scrubbing* (*Power et al., 2012*) that modeled time points with a frame-wise displacement of ≥0.5 mm, together with the preceding and subsequent time points as separate regressors during nuisance covariate correction.

We linearly co-registered the extracted hippocampal subfield volumes with the functional MRI data for each individual using FSL flirt (http://www.fmrib.ox.ac.uk/fsl/), followed by nearest neighbor interpolation. Following, we generated functional connectivity maps from both the left and right hippocampal subfields in each individual. Functional connectivity was calculated as the correlation between the mean time series of the seed region and the time series of all cortical parcels based on the Schaefer 400 parcellation. To render them normally distributed and scale the profiles across participants, correlation coefficients underwent a Fisher r-to-z transformation and were rescaled, resulting in connectivity profiles between 0 and 1 for each participant and timepoint. Functional networks were defined as the top 10% regions based on mean connectivity profile of the respective subfield in the ipsilateral hemisphere at baseline. Individuals with a framewise-displacement of >0.3 mm (<5%) were excluded.

## Diurnal cortisol assessments

For cortisol assessment, 14 saliva samples (7 per day) were obtained over the course of two consecutive weekdays (Mondays/Tuesdays, Wednesdays/Thursdays or Thursdays/Fridays, depending on participant availability). In detail, samples were taken upon free awakening (while still in bed; S1) and at 30 min, 60 min, 4, 6, 8, and 10 hr after awakening. Saliva was collected using Salivette collection devices (Sarstedt, Nuembrecht, Germany). Participants were instructed to place collection swabs in their mouths and to refrain from chewing for 2 min. They were asked to not eat, drink (except water), or brush their teeth during the 10 min before sampling, and to not smoke during the 30 min before sampling. If deviating from this guideline, they were asked to thoroughly rinse their mouth with water before taking a sample. Participants otherwise followed their normal daily routine. To maximize adherence to the sampling protocol, participants were given pre-programmed mobile devices using an in-house application that reminded them to take each (except the first) Salivette at the designated time. Sampling times of the non-morning probes were jittered (+/-15 min) to avoid complete predictability. Samples were kept in the freezer until returned to the laboratory, where they were stored at –30 °C until assay (at the Department of Biological and Clinical Psychology, University of Trier, Germany). Cortisol levels (expressed in nmol/l) were determined using a time-resolved fluorescence immunoassay (*Dressendörfer et al., 1992*) with intra-/inter-assay variability of 10/12%.

Raw cortisol data were each treated with a natural log transformation to remedy skewed distributions. Across the full sample, any values diverging more than 3 SD from the mean were labeled outliers and winsorized to the respective upper or lower 3 SD boundary to avoid influential cases. Logged and winsorized cortisol data was then averaged across the two sampling days, and the most commonly used summary indices of diurnal cortisol activity were calculated (*Ross et al., 2014*). The CAR was quantified as a change score from S1 to either the 30- or 60 min post-awakening sample, depending on the individual peak in hormone levels. If participants peaked at S1 rather than at 30 or 60 min thereafter, the 30-min data point was used to operationalize the (inverse) CAR, given that it was always closer in magnitude to S1 than the 60-min data point. The cortisol decline over the course of the day (diurnal slope) was operationalized as a change score from baseline to the final sample of the day (at 600 min after awakening). Total daily cortisol output was operationalized as the area under the curve with respect to ground, $AUC_g$ (*Pruessner et al., 2003*), which considers the difference between the measurements from each other (i.e. the change over time) and the distance of these measures from zero (i.e., the level at which the change over time occurs). Awakening, 240, 360, 480, and 600 min post-awakening cortisol values were included in the calculation of the $AUC_g$. To prevent it from having an undue influence, the CAR samples at 30 and 60 min were excluded from the total output score calculation. On each sampling day, awakening time and sleep duration were registered using the pre-programmed mobile device immediately upon awakening in parallel to taking the first Salivette. These measures were averaged across the two sampling days to minimize situational influences.

## Assay of steroid hormone concentration in hair

Please see further details on sample and dropout in *Puhlmann et al., 2021b*. To evaluate cortisol and cortisone, hair strands were taken as close as possible to the scalp from a posterior vertex position at T0 and after each following timepoint (T0-T3). Hair samples were enfolded in aluminum foil and stored in the dark at room temperature until assay at the Department of Psychology, TU Dresden, Germany. We evaluated the proximal 3 cm segment of hair to study accumulation of cortisol and cortisone over each 3-month period, based on the assumption of an average hair growth rate of 1 cm/month (*Wennig, 2000*). Hormone concentrations were captured using liquid chromatography–tandem mass spectrometry, the current criterion standard approach for hair steroid analysis (*Gao et al., 2016*). All hormone concentrations were reported in picograms per milligram. For the current longitudinal research aim, all samples of one participant were always run with the same reagent batch to avoid intraindividual variance due to batch effects.

## Quality control and case selection

Structural MRI data without artifacts and acceptable automated segmentations were available in 943 participants. Functional MRI data were available in 849 participants. We opted to have consistent sample sizes in structure and function and therefor including only people that had both structural and functional data available. Please see *Table 5* for participant numbers across timepoints and measures for structural and functional data.

Among those, salivary cortisol measures were available in *Presence* n=85 (53 females, age = 40.87 std 9.69, 20–55), *Affect* n=89 (50 females, age = 40.11 std 9.87, 20–55), *Perspective* n=81 (48 females, age = 40.14 std 9.78, 20–55). Hair cortisol change scores were available in *Presence* n=31 (21 females, age = 39.55 std 10.40, 20–54), *Affect* n=44 (24 females, age = 37.52 std 10.78, 20–54), *Perspective* n=41 (24 females, age = 38.14 std 10.51, 20–54).

## Statistical analyses

Using SurfStat (*Worsley et al., 2009*; *Larivière et al., 2023*), we carried out structural and functional MRI analysis for the left and right hippocampal subfield difference scores between different 3-month timepoints. All models statistically corrected for nuisance effects of age and sex, as well as random effect of subject. Main contrasts considered in the group analyses concern *Presence* versus *Active Control* ($T_0$-$T_1$) and *Affect* versus *Perspective* ($T_1$-$T_3$). Additionally, investigations include analyses versus Retest Control Cohort as well as subgroups defined by training cohort and timepoint. In case of multiple comparison, we performed Bonferroni correction (*Benjamini and Hochberg, 1995*).

## Partial least squares analysis

To assess potential relationships between cortisol change and hippocampal subfield volume and functional change, we performed a partial least squares analysis (PLS; *McIntosh and Lobaugh, 2004*; *Kebets et al., 2019*). PLS is a multivariate associative model that to optimizes the covariance between two matrices, by generating latent components (LCs), which are optimal linear combinations of the original matrices (*McIntosh and Lobaugh, 2004*; *Kebets et al., 2019*). In our study, we utilized PLS to analyze the relationships between change in volume and intrinsic function of hippocampal subfields and diurnal cortisol measures. Here, we included all Training Modules and regressed out effects of age, sex, and random effects of subject on the brain measures before conducting the PLS analysis. The PLS process involves data normalization within training groups, cross-covariance, and singular value decomposition. Subsequently, subfield and behavioral scores are computed, and permutation testing (1000 iterations) is conducted to evaluate the significance of each latent factor solution (FDR corrected). We report then the correlation of the individual hippocampal and cortisol markers with the latent factors. To estimate confidence intervals for these correlations, we applied a bootstrapping procedure that generated 100 samples with replacement from subjects' RSFC and behavioral data.

## Acknowledgements

Data for this project were collected between 2013 and 2016 at the former Department of Social Neuroscience at the Max Planck Institute for Human Cognitive and Brain Sciences Leipzig. We are thankful to the members of the Social Neuroscience Department involved in the ReSource Project over many years, in particular to Astrid Ackermann, Christina Bochow, Matthias Bolz and Sandra Zurborg for managing the large-scale longitudinal study, to Elisabeth Murzik, Nadine Otto, Sylvia Tydecks, and Kerstin Träger for help with recruiting and data archiving, to Henrik Grunert for technical assistance, and to Hannes Niederhausen and Torsten Kästner for data management. We wish to thank Clemens Kirschbaum for the analysis of the hair cortisol and cortisone data. Tania Singer (Principal Investigator) received funding for the ReSource Project from the European Research Council (ERC) under the European Community's Seventh Framework Program (FP7/2007–2013) ERC grant agreement number 205557. Sofie Valk received support from the Max Planck Society (Otto Hahn Award). Boris Bernhardt acknowledges research support from the NSERC (Discovery-1304413), the Canadian Institutes of Health Research (CIHR FDN-154298), SickKids Foundation (NI17-039), Azrieli Center for Autism Research (ACAR-TACC), and the Tier-2 Canada Research Chairs program. Last, we wish to thank all the participants of the study.

## Additional information

### Funding

| Funder | Grant reference number | Author |
|---|---|---|
| European Research Council | 205557 | Tania Singer |
| Natural Sciences and Engineering Research Council of Canada | Discovery-1304413 | Boris C Bernhardt |
| Canadian Institutes of Health Research | CIHR FDN-154298 | Boris C Bernhardt |
| Canadian Institutes of Health Research | CIHR PJT-174995 | Boris C Bernhardt |
| Canadian Institutes of Health Research | PJT-191853 | Boris C Bernhardt |
| Sick Kids Foundation | NI17-039 | Boris C Bernhardt |
| Azrieli Foundation | ACAR-TACC | Boris C Bernhardt |
| Canada Research Chairs | Tier 2 | Boris C Bernhardt |

| Funder | Grant reference number | Author |
| --- | --- | --- |
| Healthy Brains and Healthy Lives, Helmholtz Foundation | HIBALL | Sofie Louise Valk Boris C Bernhardt |
| Max Planck Society | Otto Hahn Award | Sofie Louise Valk |

 The funders had no role in study design, data collection and interpretation, or the decision to submit the work for publication.

## Author contributions

Sofie Louise Valk, Conceptualization, Data curation, Formal analysis, Investigation, Visualization, Methodology, Writing – original draft, Writing – review and editing; Veronika Engert, Conceptualization, Resources, Data curation, Formal analysis, Methodology, Writing – original draft, Project administration, Writing – review and editing; Lara Puhlmann, Roman Linz, Data curation, Validation, Methodology, Writing – original draft, Writing – review and editing; Benoit Caldairou, Andrea Bernasconi, Neda Bernasconi, Resources, Software, Methodology, Writing – original draft, Writing – review and editing; Boris C Bernhardt, Conceptualization, Resources, Data curation, Software, Formal analysis, Supervision, Methodology, Writing – original draft, Writing – review and editing; Tania Singer, Conceptualization, Supervision, Funding acquisition, Writing – original draft, Project administration, Writing – review and editing

## Author ORCIDs

Sofie Louise Valk ⓘ https://orcid.org/0000-0003-2998-6849
Veronika Engert ⓘ http://orcid.org/0000-0001-5317-933X
Lara Puhlmann ⓘ https://orcid.org/0000-0002-0870-8770
Boris C Bernhardt ⓘ https://orcid.org/0000-0001-9256-6041

## Ethics

Clinical trial registration #NCT01833104.
Human subjects: Human subjects: All participants gave written informed consent and the study was approved by the Research Ethics Committees of the University of Leipzig (#376/12-ff) and Humboldt University in Berlin (#2013-02, 2013-29, 2014-10).

Reviewer #1 (Public Review): https://doi.org/10.7554/eLife.87634.4.sa1
Author response https://doi.org/10.7554/eLife.87634.4.sa2

---

# Additional files

## Supplementary files

• Supplementary file 1. Supplementary analyses, descriptive statistics and supplementary tables. (a) Descriptive statistics T0-T1. (b) Descriptive statistics T1-T3. (c) T0-T1 change statistics. (d) T1-T3 change statistics. (e) T1-T3 change statistics – Training cohort 1 and 2 *Affect* versus *Perspective*. (f) T1-T2 change. (g) T2-T3 change. (h) Subfield-specific changes following the Training Modules, controlling for the other two ipsilateral subfields. (i) Overall change in subfield volume. (j) Sex differences (female versus male) in hippocampal subfield volumes. (k) Descriptive statistics mean subfield functional network change T0-T1. (l) Descriptive statistics mean subfield functional network change T1-T3. (m) Functional connectivity network change T0-T1. (n) Functional connectivity network change T1-T3. (o) Functional connectivity network change T1-T3: Training cohort 1 and 2 *Affect* versus *Perspective*. (p) Functional connectivity network change T1-T2. (q) Functional connectivity network change T2-T3. (r) Correlating change in subfield volume and diurnal cortisol indices in *Affect*. (s) Association between stress-markers and within functional network sub-regions in *Affect* and *Perspective*. (t) Correlating change in subfield functional network and diurnal cortisol indices in *Affect*. (u) Correlating change in subfield volume and diurnal cortisol indices in *Presence*. (v) Correlating change in subfield volume and diurnal cortisol indices in *Perspective*. (w) Correlating change in subfield function and diurnal cortisol indices in *Presence*. (x) Correlating change in subfield function and diurnal cortisol indices in *Perspective*. (y) Overall effects of cortisol markers on hippocampal volume in *Presence*, *Affect*, and *Perspective*. (z) Overall effects of cortisol markers

on hippocampal function in *Presence*, *Affect*, and *Perspective*. (za) Effects of hair cortisol markers on hippocampal subfield volume in *Presence*, *Affect*, and *Perspective*. (zb) Effects of hair cortisol markers on hippocampal subfield function in *Presence*, *Affect*, and *Perspective*.

• MDAR checklist

## Data availability

The present work is based on personal and sensitive biological data that could be matched to individuals. Participants did not consent to data-sharing with parties outside the MPI CBS, such that in line with the GDPR, data cannot be made publicly available. Data are available upon reasonable request (contact via valk@cbs.mpg.de). Together with the data officer at the Max Planck Institute for Human Cognitive and Brain Sciences each request will be assessed with respect to GDPR regulations. No commercial research can be performed on the data. Analysis scripts (Matlab) to reproduce primary analyses and figures are publicly available on GitHub (copy archived at *Valk, 2024*) and raw data-plots are provided whenever possible in the figures and supplementary figures.

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
