## [Editor Report · eLife assessment]

This **important** work examines the potential utility of socio-emotional and socio-cognitive mental training on hippocampal subfield structure and function, and cortisol levels. The authors provide **convincing** evidence that CA1-3 volume is sensitive to socio-emotional training, with changes related to function plasticity and cortisol levels. Further, the authors provide evidence of change across all subfields and training modules related to stress.

---

## [Referee Report · Reviewer #1 (Public Review)]

Valk and Engert et al. examined the potential relations between three different mental training modules, hippocampal structure and functional connectivity, and cortisol levels (stress) over a 9-month period. They found that among the three types of mental training: Presence (attention and introspective awareness), Affect (socio-emotional - compassion and prosocial motivation), and Perspective (socio-cognitive - metacognition and perspective taking) modules; Affect training most robustly related to changes in hippocampal structure and function - specifically, CA1-3 subfields of the hippocampus. Moreover, change in intrinsic functional connectivity related to changes in diurnal cortisol release and long-term cortisol exposure. These changes are proposed to result from a combination of factors, which is supported by multivariate analyses showing changes across subfields and training content relate to cortisol changes.

The authors demonstrate that mindfulness training programs are a potential avenue for stress interventions that impact hippocampal structure and cortisol, providing a promising approach to improve health. The data contribute to the literature on plasticity of hippocampal subfields during adulthood, the impact of mental training interventions on the brain, and the link between CA1-3 and both short- and long-term stress changes.

The authors thoughtfully approached the study of hippocampal subfields, utilizing a method designed for T1w images that outperformed Freesurfer 5.3 and that produced comparable results to an earlier version of ASHS. The authors note the limitations of their approaches and provide detailed information on the data used and analyses conducted. The results provide a strong basis from which future studies can expand using computational approaches or more fine-grained investigations of the impact of mindfulness training on cortisol levels and the hippocampus.

---

## [Author Response]

The following is the authors’ response to the previous reviews.

**Public Reviews:**

**Reviewer #1 (Public Review):**
Valk and Engert et al. examined the potential relations between three different mental training modules, hippocampal structure and functional connectivity, and cortisol levels (stress) over a 9-month period. They found that among the three types of mental training: Presence (attention and introspective awareness), Affect (socio-emotional - compassion and prosocial motivation), and Perspective (socio-cognitive - metacognition and perspective taking) modules; Affect training most robustly related to changes in hippocampal structure and function - specifically, CA1-3 subfields of the hippocampus. Moreover, change in intrinsic functional connectivity related to changes in diurnal cortisol release and long-term cortisol exposure. These changes are proposed to result from a combination of factors, which is supported by multivariate analyses showing changes across subfields and training content relate to cortisol changes.The authors demonstrate that mindfulness training programs are a potential avenue for stress interventions that impact hippocampal structure and cortisol, providing a promising approach to improve health. The data contribute to the literature on plasticity of hippocampal subfields during adulthood, the impact of mental training interventions on the brain, and the link between CA1-3 and both short- and long-term stress changes.The authors thoughtfully approached the study of hippocampal subfields, utilizing a method designed for T1w images that outperformed Freesurfer 5.3 and that produced comparable results to an earlier version of ASHS. The authors note the limitations of their approaches and provide detailed information on the data used and analyses conducted. The results provide a strong basis from which future studies can expand using computational approaches or more fine-grained investigations of the impact of mindfulness training on cortisol levels and the hippocampus.

We thank the Reviewer for the positive re-evaluation and summary of our findings and work. We made additional change as suggested and hope this clarified any open points.

I have a few additional suggestions. Clarifying the language around the multivariate results and the impact across subfields and training modules would be helpful.

We are happy to provide further clarifications with respect to the multivariate results and the impact of training on subfields.

The multivariate analyses served as a final step to explore any potential connections between training modules and hippocampal subfields, beyond just the link between CA1-3 and the *Affect* Module. These additional analyses were suggested by the Reviewers, and we, as authors, agreed that taking a broader view of how different parts of the hippocampus interact with overall changes can provide valuable insights into the relationship between mental training, cortisol fluctuations, and changes in CA1-3 subfields.

We employed a multivariate partial least squares method, which aims to identify the directions in the predictor space that account for the most variance in changes observed, by creating latent variables. Initially, we investigated whether there was a general connection between CA1-3 subfields and cortisol changes, regardless of which training module produced these effects. Our findings confirmed a consistent relationship across all three training modules, indicating a strong association between cortisol changes, particularly markers such as AUC and slope change, and alterations in CA1-3 structure and functional connectivity. We explored a model incorporating changes across all hippocampal subfields and stress markers across different modules. In the right hemisphere, changes in the volume of the CA1-3 subfield were more strongly associated with stress markers, compared to other subfields. However, this association was less pronounced in the left hemisphere.

Our multivariate approach captured fluctuations across subfields and modules beyond group-level associations, leading to a more nuanced interpretation. While the univariate analysis of module-specific changes in volume and associations within the *Affect* Module may offer a straightforward interpretation, as they coincide with increases in CA1-3 volume, the multivariate analysis also accounts for individual-level changes not observed at the group level using a data-driven approach. Overall these findings are in line with the group-level observations, yet provide nuance on specificity.

We clarified these considerations further in the manuscript;

Abstract:

“Notably, using a multivariate approach, we found that other subfields that did not show group-level changes also contributed to changes in cortisol levels.”

Results:

“We employed a multivariate partial least squares method, which aims to identify the directions in the predictor space that account for the most variance in changes observed, by creating latent variables. Initially, we investigated whether there was a general connection between CA1-3 subfields and cortisol changes, regardless of which training module produced these effects.”

Discussion:

“Finally, through conducting multivariate analysis, we once more noticed associations between changes in CA1-3 volume and functional adaptability and alterations in stress levels, particularly prominent within the Affect Module. Integrating all subfields into a unified model highlighted a distinct significance of CA1-3, although for the left hemisphere, we observed a more diverse range of contributions across subfields. In summary, we establish a connection between a socio-emotional behavioral intervention, shifts in hippocampal subfield structure and function, and decreases in cortisol levels among healthy adults.

Although the univariate examination of changes specific to modules in volume and connections within the Affect Module presents how changes in cortisol align with group-level rises in CA1-3 volume, the multivariate analysis extended this observation through considering individual-level alterations not discernible at the group level through a data-driven method. These results generally corresponded with observations at the group level but offer additional insights into specificity, and hint at system-level alterations.”